# Targetable *NOTCH1* rearrangements in reninoma

Taryn D. Treger[1,2,3,12], John E. G. Lawrence[1,3,12], Nathaniel D. Anderson[1], Tim H. H. Coorens [4], Aleksandra Letunovska[5,6], Emilie Abby [1], Henry Lee-Six [1,7], Thomas R. W. Oliver [1,7], Reem Al-Saadi [5,6], Kjell Tullus[5,6], Guillaume Morcrette[5,6], J. Ciaran Hutchinson[6], Dyanne Rampling[6], Neil Sebire[5,6], Kathy Pritchard-Jones[5], Matthew D. Young [1], Thomas J. Mitchell [1,3,8], Philip H. Jones [1,9], Maxine Tran[10,11,13] ✉, Sam Behjati [1,2,3,13] ✉ & Tanzina Chowdhury[5,6,13] ✉

Reninomas are exceedingly rare renin-secreting kidney tumours that derive from juxtaglomerular cells, specialised smooth muscle cells that reside at the vascular inlet of glomeruli. They are the central component of the juxtaglomerular apparatus which controls systemic blood pressure through the secretion of renin. We assess somatic changes in reninoma and find structural variants that generate canonical activating rearrangements of, *NOTCH1* whilst removing its negative regulator, *NRARP*. Accordingly, in single reninoma nuclei we observe excessive renin and NOTCH1 signalling mRNAs, with a concomitant non-excess of *NRARP* expression. Re-analysis of previously published reninoma bulk transcriptomes further corroborates our observation of dysregulated Notch pathway signalling in reninoma. Our findings reveal *NOTCH1* rearrangements in reninoma, therapeutically targetable through existing NOTCH1 inhibitors, and indicate that unscheduled Notch signalling may be a disease-defining feature of reninoma.

Efforts of the past decade characterising the genomes of human cancer have shown that the rarest tumour types are often underpinned by highly recurrent somatic events[1]. Such pathognomonic mutations are, occasionally, also targetable through established pharmacological agents. The discovery of somatic changes that define rare tumours may therefore have immediate clinical diagnostic and therapeutic utility.

Amongst the rarest of tumours in humans are reninomas (juxtaglomerular cell tumours), with only approximately 100 cases reported to date[2]. Reninomas represent a neoplastic expansion of juxtaglomerular cells of the kidney, cells that regulate blood pressure via the secretion of renin. Consequently, reninomas tend to be detected through imaging performed in patients with intractable hypertension. It is this severe hypertension that accounts for the principal clinical challenge associated with these tumours. Furthermore, both local recurrences and distant metastases have been described[3]. Treatment involves aggressive control of hypertension followed by surgical resection, where possible, by nephron-sparing surgery. There are no

[1]Wellcome Sanger Institute, Hinxton CB10 1SA, UK. [2]Department of Paediatrics, University of Cambridge, Cambridge CB2 0QQ, UK. [3]Cambridge University Hospitals NHS Foundation Trust, Cambridge CB2 0QQ, UK. [4]Broad Institute of MIT and Harvard, Cambridge 02142 MA, USA. [5]UCL Great Ormond Street Institute of Child Health, London WC1N 1EH, UK. [6]NIHR Great Ormond Street Hospital Biomedical Research Centre, London WC1N 3JH, UK. [7]Department of Pathology, Cambridge University Hospitals NHS Foundation Trust, Cambridge CB2 0QQ, UK. [8]Early Cancer Institute, University of Cambridge, Cambridge CB2 0XZ, UK. [9]Department of Oncology, University of Cambridge, Cambridge CB2 0XZ, UK. [10]Specialist Centre for Kidney Cancer, Royal Free Hospital, London NW3 2QG, UK. [11]Faculty of Medical Sciences, Division of Surgery and Interventional Science, University College London, London NW3 2PS, UK. [12]These authors contributed equally: Taryn D. Treger, John E. G. Lawrence. [13]These authors jointly supervised this work: Maxine Tran, Sam Behjati, Tanzina Chowdhury. ✉e-mail: m.tran@ucl.ac.uk; sb31@sanger.ac.uk; Tanzina.Chowdhury@gosh.nhs.uk

medical therapeutics that target reninoma, and the genetic drivers underpinning this tumour are unknown. Studies of gene expression in reninoma have described several pathways as dysregulated in bulk tumour transcriptomes yet have not identified an overarching aberration[4].

Investigations into the regulation of renin secretion by juxtaglomerular cells have proposed a central role for NOTCH1 signalling[5–8]. For example, gene knockout of *Rbpj*, a key effector of NOTCH1 signalling, decreased the number of juxtaglomerular cells and renin secretion by individual cells[6]. Moreover, *NOTCH1* operates as a dominant cancer gene (oncogene) in a variety of neoplasms through mutations encompassing rearrangements, point mutations and indels[9,10]. These mutations constitutively activate NOTCH1 signalling by removing the intracellular signalling domain from control of the extracellular inhibitory domain. Although *NOTCH1* has not been directly implicated as a driver of reninoma, somatic fusion events in *NOTCH1* and its paralogues, *NOTCH2* and *NOTCH3*, have been reported to underpin glomus tumours, which are neoplasms arising from glomus bodies (typically in nail beds) and which are histological mimics of reninoma[11].

Here, we examine two reninomas by whole genome and bulk RNA sequencing and find a canonical *NOTCH1* rearrangement in both which, in conjunction with single nuclear mRNA data and re-analyses of previously published bulk transcriptomes, indicates that unscheduled Notch signalling may drive reninoma.

## Results

### Canonical *NOTCH1* rearrangements in two reninomas

We investigated, initially by whole genome sequencing (WGS), tumours of two individuals with reninoma; a child (case 1, PD50642) with a localised tumour and a young adult (case 2, PD54845) with metastatic (lung) reninoma at presentation. We examined the genomes of primary tumours (*n* = 2 samples from case 1; *n* = 1 sample from case 2), metastasis (case 2), normal kidney (both cases) and blood cell-derived DNA (case 1). We called all classes of somatic variation—substitutions, indels, rearrangements and copy number variation—using an established variant calling pipeline[12–15], using blood (case 1) or normal kidney (case 2) to subtract germline variants. A genome-wide overview of these somatic changes is provided in Supplementary Fig. 1. Most somatic features were unremarkable, apart from multiple copy number gains and losses in case 1, consistent with karyotypic findings previously described in reninomas[16]. Our key finding was a 0.8 MB deletion on chromosome 9q seen in the tumours of both patients. In case 1, this was caused by a single deletion event, and in case 2, by a more complex configuration (Fig. 1a) comprising an intragenic *NOTCH1* inversion between intron 27 and intron 28, in addition to the aforementioned deletion. Annotation of breakpoints at the 5' end revealed that *NOTCH1* was truncated within the regulatory region of the extracellular domain in both cases, akin to activating rearrangements that have been described in T-cell leukaemia (Fig. 1b, c)[9]. Such rearrangements constitutively activate NOTCH1 by removing the regulatory region and releasing the intracellular domain that mediates the signalling cascade of NOTCH1. Importantly, in case 1, the breakpoint spared the transmembrane region (encoded by exon 28), which can be targeted through established γ-secretase inhibitors, that act to prevent cleavage of the intracellular domain[17,18]. In case 2, targeting the downstream NOTCH1 transcription activation complex may be more appropriate as the effect of the intragenic *NOTCH1* inversion on γ-secretase binding is unknown[19]. At the 3' end of the deletions, the breakpoints were, most unusually, in close proximity in both cases (3 Kb). In the paediatric case, the 3' breakpoint occurred within the promoter of NOTCH Regulated Ankyrin Repeat Protein gene (*NRARP*), whilst in case 2, the entire *NRARP* gene was encompassed within the deletion (Fig. 1a). The functional importance of NRARP is that it is a negative regulator of NOTCH1 signalling[20,21]. Overall, both cases

harboured rearrangements that generated an activating truncation of the *NOTCH1* gene whilst removing one copy of the NOTCH1 inhibitor, *NRARP*.

### Validation of *NOTCH1* rearrangements

To validate *NOTCH1* rearrangements, we performed RNA sequencing of all tumours. This confirmed expression of *NOTCH1* exons that encode the intracellular (signalling) and the transmembrane domains whilst omitting exons of the extracellular (regulatory) domain (Fig. 2a). Accordingly, domain-specific expression of *NOTCH1* exons showed a stark excess of mRNA encoding the intracellular domain over the extracellular domain compared to transcription of the wildtype transcript in different tissues (Fig. 2b). Consistent with this finding, immunohistochemistry of tumour specimens with an antibody targeting NOTCH1 at the C terminus (Ab52627) demonstrated fully nuclear staining (Fig. 2c). Similarly, immunofluorescence of reninoma further confirmed overexpression of NOTCH1 intracellular domain and renin by the same cells (Fig. 2d). Additionally, detection of the NOTCH1 intracellular domain using a specific epitope at the S3 cleavage site (D3B8 antibody) supports its release by γ-secretase cleavage in case 1 (Fig. 2d).

### Transcriptional consequences of the *NOTCH1* rearrangement

To study transcriptional consequences of the *NOTCH1* rearrangement, we generated single nuclei mRNA data derived from fresh frozen tissues of primary tumours from both individuals (Chromium 10×), using standard protocols[22]. Applying stringent quality filtering, we obtained readouts from a total of 29,835 nuclei (case 1 = 15,448 nuclei; case 2 = 14,387 nuclei) (Supplementary Fig. 2). Amongst these, we were able to discern tumour cells by applying previously reported markers[4] of reninoma which delineated a group of renin producing cells, co-expressing *NOTCH1* and *NRARP* (Fig. 3a) (Supplementary Fig. 2). Based on single nuclei readouts, we quantified the expression of relevant genes relative to normal human mesangial-like single-cell transcriptomes, identified within a recently published human single renal cell data set[23], annotated through published markers of murine mesangial-like cells[24] (Supplementary Fig. 3).

Examining these data, first, we asked whether each tumour cell transcribed more renin than normal mesangial-like cells. Comparing normalised expression values, we found a significant increase in renin expression in tumours compared to normal human mesangial-like cells (Wilcoxon rank-sum test, *p* < 0.05; Fig. 3b). This would indicate that the excess of renin production by reninomas may not only result from an increased number of renin producing cells but also from elevated renin transcription within each cell. We next queried whether NOTCH1 signalling was dysregulated within tumour cells, especially given the genomic disruption in each tumour of one allele of *NRARP*, the negative regulator of NOTCH1. To this end, we compared the ratio of the canonical effector molecules of NOTCH1 that were expressed in the tumours (*HEYL*, *HEY1*, *HEY2*, *HES1*, *HES4*, *HES5*)[25] over *NRARP*. We found a significant increase (Wilcoxon rank-sum test; *p*-values detailed in figure) in tumour versus normal cells for all targets in both tumours, with the exception of *HES4* in nuclei derived from one of the tumours (Fig. 3c). These findings indicate a non-excess of *NRARP* mRNAs in the presence of an activating *NOTCH1* rearrangement.

### Evidence of NOTCH1 activity in published reninoma transcriptomes

*Martini* and colleagues have previously published expression analyses of four reninomas[4], alongside one normal kidney sample, the raw sequencing reads of which we obtained for re-analyses. Following the remapping of reads, we compared expression values of individual genes across analyses and found an essentially perfect correlation (Supplementary Fig. 4), ruling out technical, analytical differences as a source of findings.

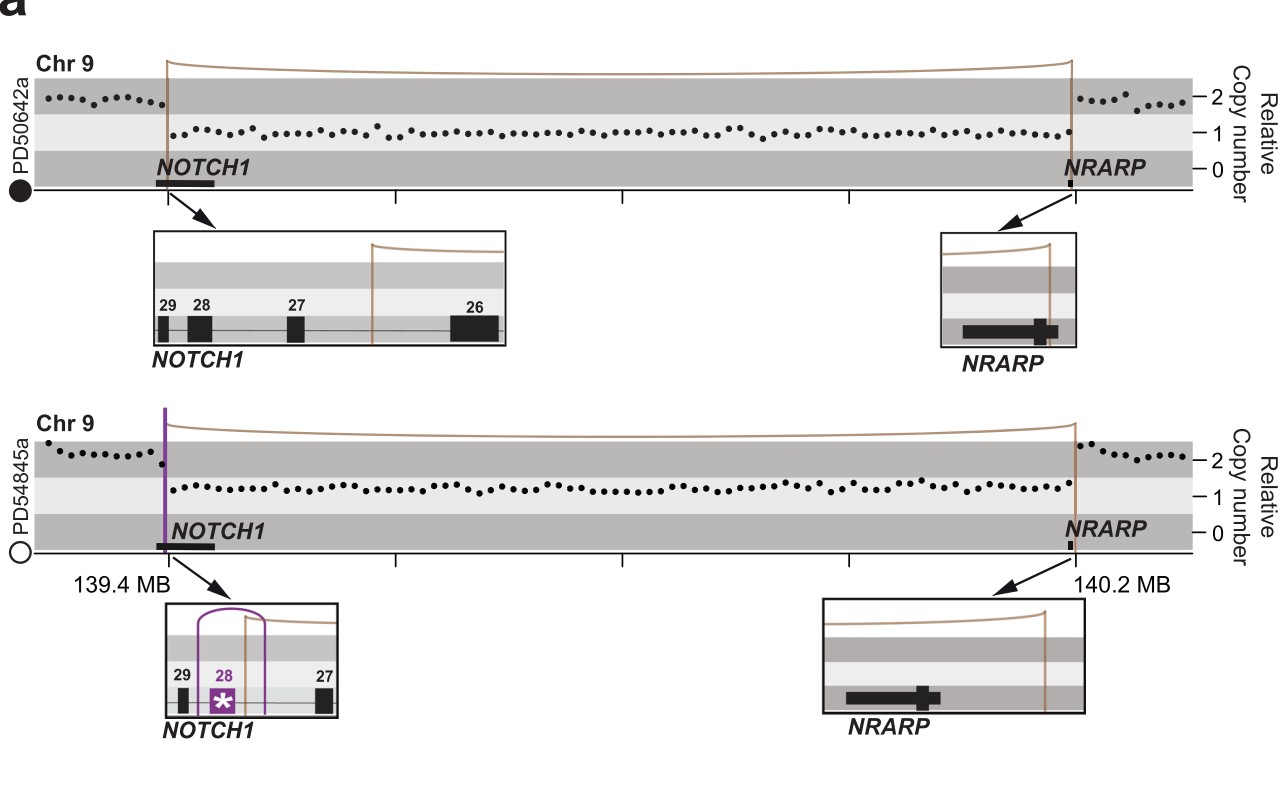

**b**

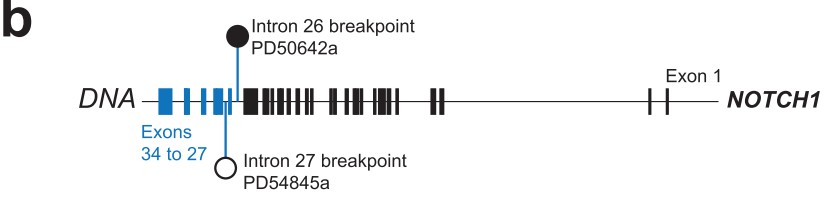

**c**

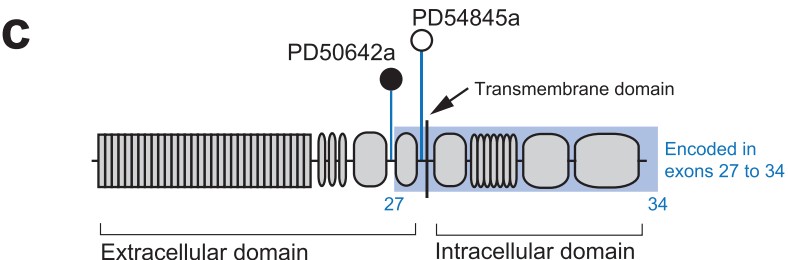

**Fig. 1 | Driver events involving *NOTCH1* and *NRARP* in two reninoma cases.**
**a** Rearrangement plots showing copy number, rearrangements and position on chromosome 9. Boxes detail the position of breakpoints in relation to genes. * highlights exon 28 involved in the inversion in PD54845. **b** *NOTCH1* gene, illustrating where breakpoints occur in each case. **c** NOTCH1 schematic, illustrating breakpoints occurring in the regulatory region in each case.

We examined the overlap in differential gene expression between reninoma data sets. We determined differential gene expression analyses on each set of tumours independently (relative to corresponding normal tissue) and intersected gene lists. Amongst shared differentially expressed genes were targets of Notch signalling, *NOTCH1* itself and *NRARP* (Fig. 4a, b). This would suggest that prominent Notch signalling was a common feature across tumour sets. We then looked for evidence of fusion events in *NOTCH1* and its paralogues using algorithmic approaches, as well as manual inspection of exon coverage and split reads. None of these approaches revealed a fusion event. It is possible that driver events in these four tumours are point mutations in *NOTCH1* or in other genes that dysregulate NOTCH1 signalling, which we cannot unequivocally assess in bulk transcriptomes. Finally, we compared *NOTCH1* expression between both reninoma data sets and bulk transcriptomes from congenital mesoblastic nephroma ($n = 21$), Wilms tumour ($n = 308$), renal cell carcinoma (RCC, $n = 824$) and normal kidneys ($n = 332$) and found significant overexpression of

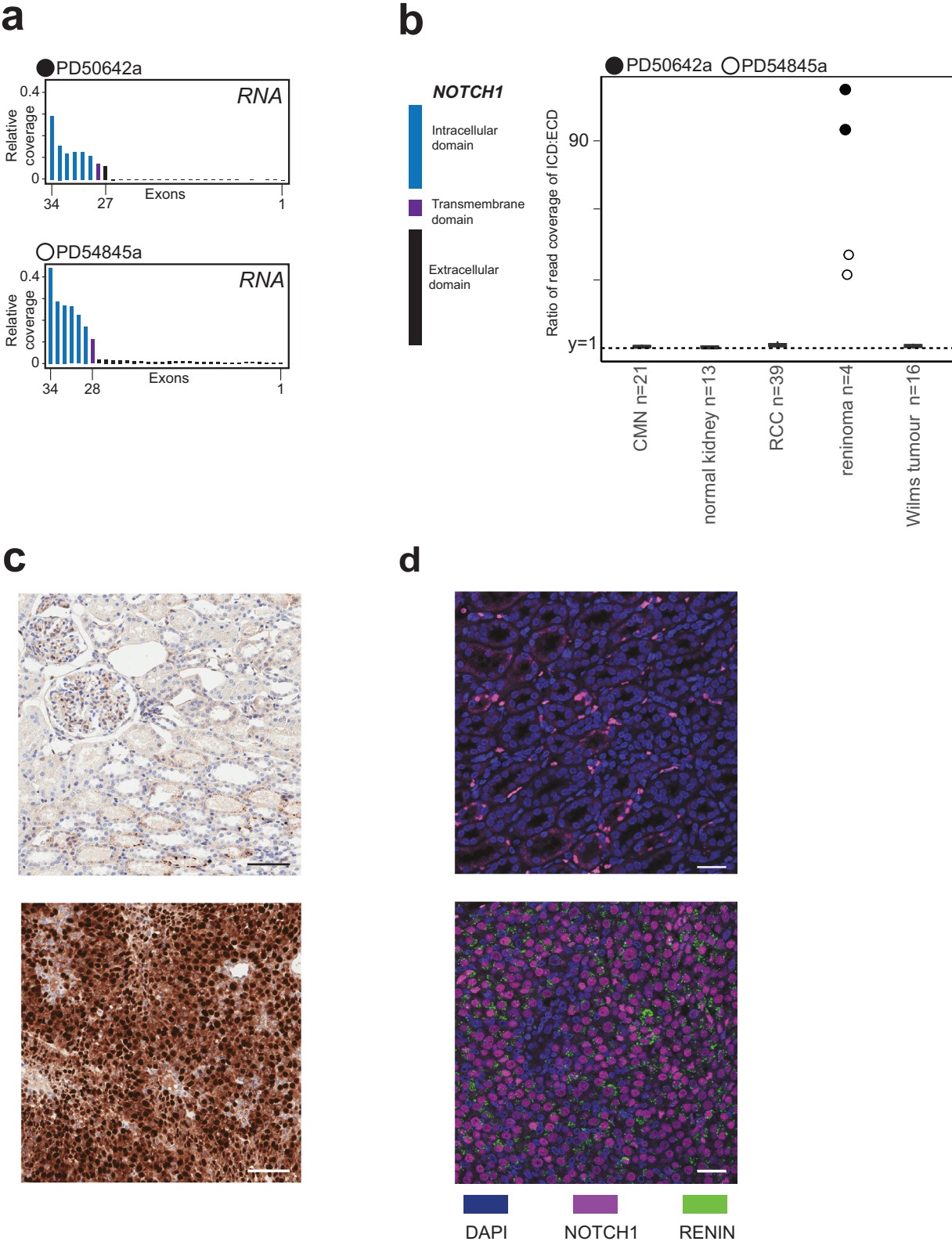

**Fig. 2 | Expression and staining of *NOTCH1* in reninomas. a** Normalised coverage of *NOTCH1* from RNA sequencing, plotted per exon, for both primary reninomas. Transmembrane domain highlighted. **b** Ratio of normalised coverage of *NOTCH1* intracellular signalling domain vs. extracellular domain, plotted for reninomas, congenital mesoblastic nephroma (CMN, *n* = 21), Wilms tumour (*n* = 16), normal kidney (*n* = 13) and renal cell carcinoma (RCC, *n* = 39). **c** Immunohistochemistry of normal kidney (upper panel) and tumour from case 1 (lower panel) with an antibody targeting NOTCH1 at the C terminus (Ab52627). The experiment was repeated 3 times. Scale bars 100 μm, original magnification 6.5×. **d** Immunofluorescence with antibodies targeting activated NOTCH1 intracellular domain (D3B8) and renin(ab134783). The upper panel shows staining in the normal kidney. The lower panel highlights the co-expression of activated NOTCH1 (purple staining) and renin(green staining) in reninoma from case 1. The experiment was performed once. Scale bars 30 μm, original magnification 40×.

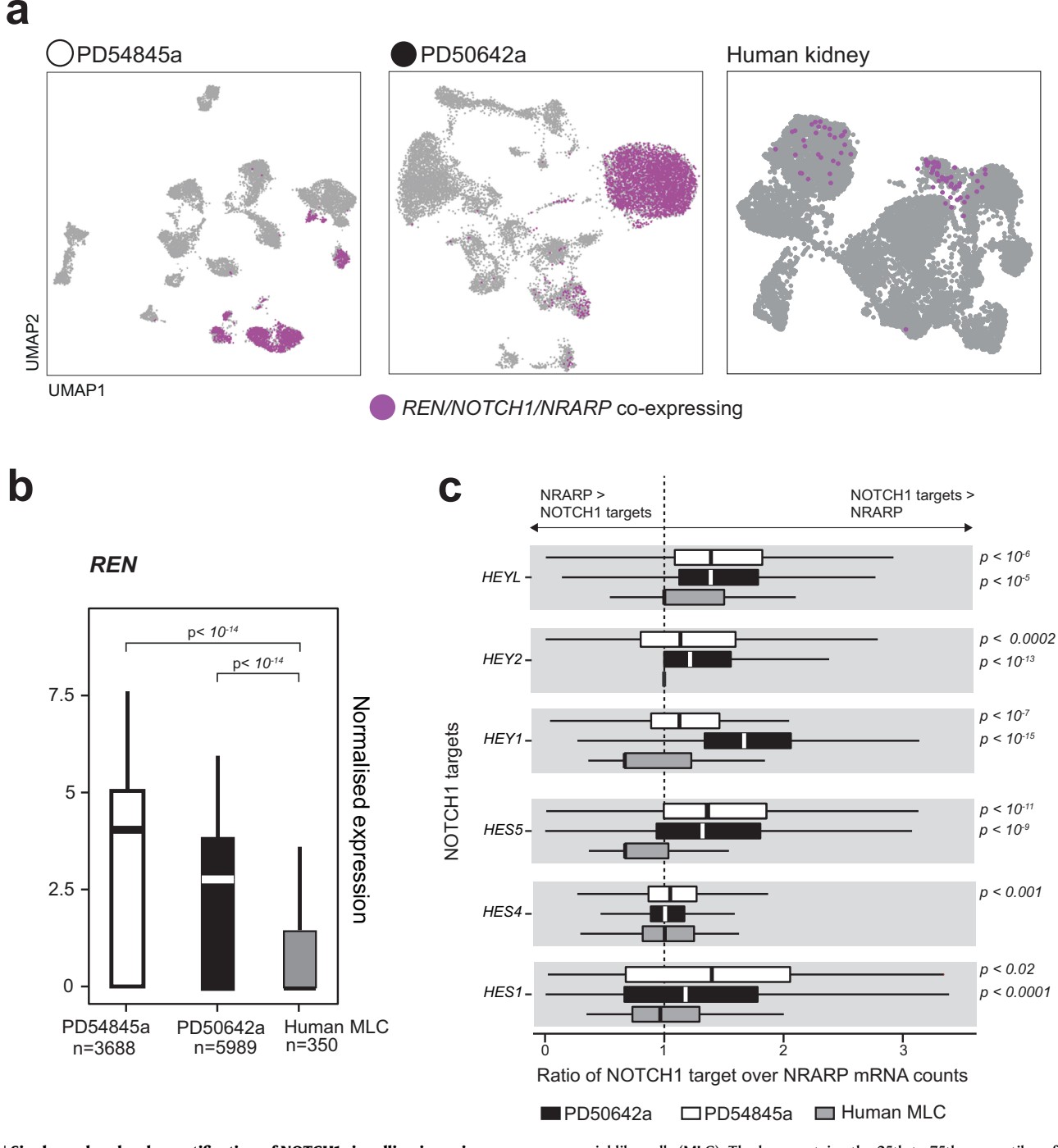

**Fig. 3 | Single nuclear level quantification of NOTCH1 signalling in reninomas and in mesangial-like cells. a** Uniform Manifold Approximation and Projection (UMAP) plots of tumour (left and middle panels) and healthy human kidney (right panel) cells. **b** Box plot quantifying renin expression in tumour cells and mesangial-like cells (MLC). The box contains the 25th to 75th percentiles of the data, with the central line denoting the median value. The upper whisker extends from the median to the largest value, no further than the 1.5 * inter-quartile range (IQR). The lower whisker extends from the median to the smallest value, at most 1.5 * IQR. Wilcoxon rank-sum test, two-sided, with adjustment for multiple comparisons. **c** Box plots showing the ratio of *NOTCH1* target genes vs. *NRARP* for tumours and mesangial-like cells (MLC). The box contains the 25th to 75th percentiles of the data, with the central line denoting the median value. The upper whisker extends from the median to the largest value, no further than the 1.5 * inter-quartile range (IQR). The lower whisker extends from the median to the smallest value, at most 1.5 * IQR. Wilcoxon rank-sum test, two-sided, with adjustment for multiple comparisons. HEYL: PD50642a $n = 1569$, PD54845a $n = 2517$, MLC $n = 111$; HEY2: PD50642a $n = 1164$, PD54845a $n = 2913$, MLC $n = 104$; HEY1: PD50642a $n = 185$, PD54845a $n = 536$, MLC $n = 111$; HES5: PD50642a $n = 706$, PD54845a $n = 1171$, MLC $n = 52$; HES4: PD50642a $n = 3174$, PD54845a $n = 5828$, MLC $n = 144$; HES1: PD50642a $n = 28$, PD54845a $n = 517$, MLC $n = 128$. Source data are provided as a Source Data file.

*NOTCH1* in reninomas[26–30] (Wilcoxon rank-sum test, $p < 0.05$; Fig. 4c). Single-cell data corroborated this finding, with significant increase in *NOTCH1* expression in each tumour cell compared to normal human mesangial-like cells (Wilcoxon rank-sum test, $p < 0.05$; Fig. 4d).

## Discussion

Despite the disparate clinical phenotypes of localised vs. metastatic disease, we found a unifying somatic change in two reninomas; classical activating *NOTCH1* rearrangements that have been previously described as driver events, corroborated by transcriptional and

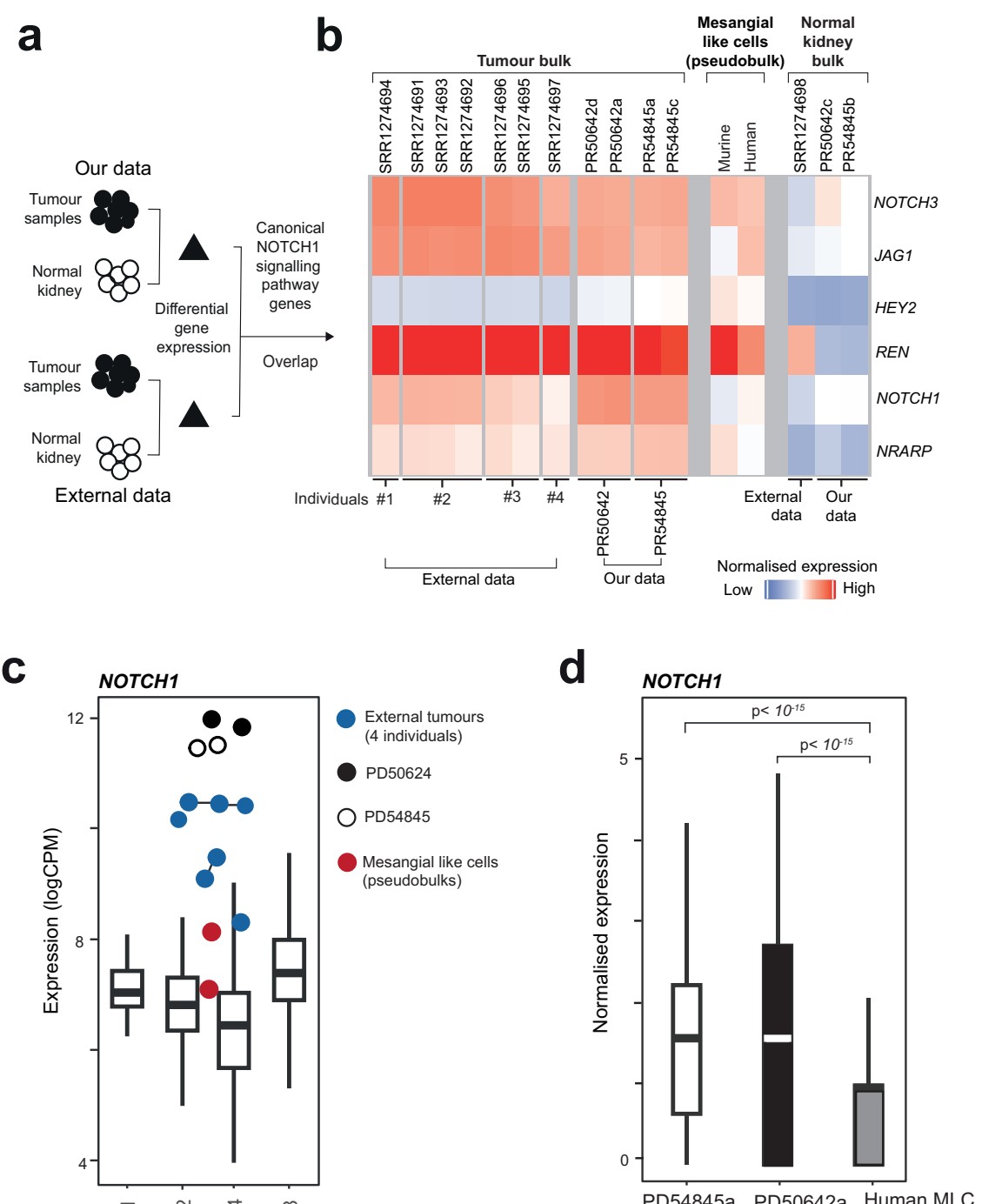

protein level evidence. Cancer cells often hijack pre-existing physiological pathways that operate in their cell of origin. Given the established role of *NOTCH1* in the physiology of juxtaglomerular cells[6], *NOTCH1* would seem to be a plausible driver of reninoma.

As an exceedingly rare tumour, an invariable shortcoming of our study is that we are unable to access additional fresh frozen samples of reninoma. Re-analysis of bulk transcriptomes of four previously published reninomas[4] supports a role for dysregulated Notch signalling in

these tumours, despite the underlying somatic mutation evading detection by RNA sequencing. It is conceivable that non-rearrangement variants or mutations in functionally related genes operate in these four tumours. A relevant precedent in this context is driver variants in glomus tumours. These neoplasms are morphological mimics of reninoma that typically arise in nail beds from glomus body cells, which are specialised vascular cells related to juxtaglomerular cells[31]. Glomus tumours harbour driver events in *NOTCH1*, its

**Fig. 4 | Notch signalling across in-house and existing bulk reninoma transcriptomes. a** Schematic of differential gene expression analyses across different data sets. **b** Heatmap showing expression of Notch signalling genes and *REN* across reninoma, matched normal kidney, human mesangial-like cells and murine mesangial-like cells. **c** *NOTCH1* expression (log-CPM (counts per million)) in reninomas, compared to mesangial-like cells (psuedobulks, *n* = 2), normal kidney (*n* = 332), congenital mesoblastic nephroma (CMN, *n* = 21), Wilms tumour (*n* = 308) and renal cell carcinoma (RCC, *n* = 824). The box contains the 25th to 75th percentiles of the data, with the central line denoting the median value. The upper

whisker extends from the median to the largest value, no further than the 1.5 * inter-quartile range (IQR). The lower whisker extends from the median to the smallest value, at most 1.5 * IQR. **d** Box plot quantifying *NOTCH1* expression in tumour cells and mesangial-like cells (MLC). The box contains the 25th to 75th percentiles of the data, with the central line denoting the median value. The upper whisker extends from the median to the largest value, no further than the 1.5 * inter-quartile range (IQR). The lower whisker extends from the median to the smallest value, at most 1.5 * IQR. Wilcoxon rank-sum test, two-sided, with adjustment for multiple comparisons. Source data are provided as a Source Data file.

paralogues and other cancer genes that functionally converge in aberrant Notch signalling[11].

Our findings have immediate clinical relevance. Reninoma is primarily a surgical disease, even in cases of metastases. However, occasionally surgery will be contra-indicated or technically unfeasible. The *NOTCH1* rearrangement we describe here would be considered a targetable mutation. As reninomas do not respond to conventional anti-cancer therapies, it may be reasonable therefore, to consider NOTCH1 inhibitors in patients with otherwise incurable reninoma. Interestingly, using such a precision oncology approach, a child with a *NOTCH1* rearranged metastatic glomus tumour was successfully treated with NOTCH1 inhibitors[32].

As systematic efforts to study cancer genomes have largely concluded, opportunistic investigations of rare tumour types, as we have performed here in reninoma, are required to complete the compendium of the human cancer genome.

## Methods

### Patients and sampling
Case 1 (PD50642) was enrolled in the UMBRELLA study, which was approved by the national research ethics committee (London Bridge Research Ethics Committee 12/LO/0101). *Case 2* (PD54845) was enrolled in the Characterisation of the immunological and biological markers of Renal cancer progression (West of Scotland Research Ethics Committee 16/WS/0039). All patients registered (or their legal guardians) provided signed informed consent to undertake genetic testing of their samples. The sex of the patients was recorded, but no further analysis was carried out due to the limited number of patients. Owing to the rarity of the tumours, no statistical method was used to predetermine sample size, and no data were excluded from the analyses.

For both cases, normal kidney and tumour tissues were collected at surgery. Normal kidney was sampled by pathologists, according to the study protocols from a morphologically normal appearing corticomedullary region distant from the tumour.

### Nucleic acid extraction
Samples used for DNA/RNA extraction were fresh-frozen and stored at −80 °C. Tumour and normal kidney DNA/RNA were extracted by standard methods using the Qiagen Allprep kit/QIAmp DNA kit/RNeasy kit. Blood germline DNA was extracted with the QIAamp DNA Blood Midi Kit (Qiagen).

### DNA sequencing and alignment
Short insert (500 bp) genomic libraries were constructed, flowcells prepared and 150 bp paired-end sequencing clusters generated on the Illumina HiSeq X platform. Tumours were sequenced to 60–100×, normal kidney to 90× and blood to 60×. DNA sequences were aligned to the GRCh37 reference genome by the Burrows–Wheeler algorithm (BWA-MEM)[33].

### Detection of variants
Using the extensively validated analysis pipeline of the Wellcome Sanger Institute, all classes of mutations were called: substitutions (CaVEMan), indels (Pindel), copy number variation (ASCAT and Battenberg) and rearrangements (BRASS)[12–15]. The CaVEMan, Pindel and

BRASS algorithms were run matched (tumour versus matched normal) and unmatched (all samples against an in silico human reference genome). Mapping artefacts were removed by setting a threshold for the median alignment score of reads supporting a variant (ASMD ≥ 140) and requiring that fewer than half of the reads were clipped (CLPM = 0). Pindel variants were filtered out if the quality score was below 300. Rearrangements were validated if they met the criteria of either an assembly score or a minimum number of reads (4 in tumours, 25 in normal samples). Despite running two copy number algorithms, no consistent solution could be found for case 1 due to a very low tumour purity (Supplementary Fig. 1).

### Annotation of somatic driver events
Variants in genes considered cancer genes were annotated, as per Census of Cancer Genes[34]. Missense mutations and in-frame indels were considered drivers if occurring in canonical hot spots of oncogenes. Truncating mutations were considered drivers if predicted to disrupt the footprint of recessive cancer genes. Focal (<1 MB) homozygous deletions and amplifications (copy number >4 (diploid) or 8 (tetraploid)) in recessive and dominant cancer genes, respectively, were considered drivers. Rearrangements were considered driver events when they generated a known oncogenic gene fusion or when their breakpoints disrupted the gene footprint of recessive genes. Rearrangements were validated by manual inspection of both WGS and RNA sequencing data on the genome browser JBrowse to exclude further sequencing artefacts[35].

### Bulk RNA analysis
RNA libraries were sequenced on the Illumina HiSeq 4000 platform. Reads were aligned using STAR and mapped to GRCh37, and read counts of genes were obtained using HTSeq[36,37]. Data was processed in R using Edge R, normalised using the TMM method and converted to log-CPM values[38]. Differential expression analysis was performed using Limma and Glimma[39,40]. Multiple hypothesis correction testing was performed on *p*-values of differentially expressed genes using the Benjamini–Hochberg method[41]. Bwcat (version 1.5.2), from the in-house cgpBigWig package, was used to determine raw coverage across genes of interest and then normalised per exon prior to plotting using ggplot2 in R[42]. The ratio of read coverage of the intracellular signalling domain (exons 29-34) to the extracellular domain (exons 1–27) was used to determine domain-specific expression.

### Immunohistochemistry
Renal tissue from the tumour and normal kidney for case 1 was stained with the Ab52627 Abcam antibody, including 1:150 and 1:300 concentrations (Supplementary Data 4). The tissue was fixed in 10% buffered neutral formalin for 24 h and processed on Leica-Peloris Tissue Processor before being embedded in wax. Sections were cut on the microtome at 4 μm and placed on the charged slides. The slides were then dewaxed and rehydrated using a Bond Max machine with epitope retrieval heat-induced in citrate buffer, pH 6.0, for 30 min at 60 °C. Staining was performed using Leica-BOND Polymer Refine Detection kit, including peroxidase block- DS9800. The tissue was counterstained with haematoxylin, dehydrated and mounted with DPX.

## Immunofluorescence

For immunofluorescence staining, dewaxed and rehydrated tissue sections were treated for antigen retrieval by boiling for 30 min in citrate buffer (pH 6). Sections were then blocked in staining buffer (10% donkey serum, 0.5% bovine serum albumin, 0.25% fish skin gelatin, 0.5% Triton X-100 in PBS) for 30 min and incubated overnight in the respective primary antibodies diluted in staining buffer (Supplementary Data 4). Secondary antibody incubations lasted for 3 h. Slides were washed three times in PBS between incubations. Finally, nuclei were counterstained by incubation for 30 min in $0.5\,\mu g\,ml^{-1}$ 4,6-diamidino-2-phenylindole (DAPI)/PBS solution and slides were mounted using Vectashield mounting media (Vector Laboratories, reference: H-1000). Sections were imaged using a Leica TCS SP8 confocal microscope at ×40 objective. Acquisition settings were optimal pinhole, line average 4 and scan speed of 400 Hz. The resolution was 1024 × 1024 pixels. Velocity 6 software was used for image visualisation.

## Tumour processing for single nuclear RNA sequencing

Frozen tissue samples were cut into $1\,mm^2$ fragments prior to transfer in homogenisation buffer to a loose dounce homogeniser for initial dissociation through twenty strokes. A second tight dounce homogeniser was then swapped in for a further 20 strokes, followed by visual confirmation that the tissue had dissociated. Filtration through a 40 µM cell strainer on ice was performed to ensure no larger fragments remained, and the remaining nuclei were visually inspected under a microscope to ensure no clumping. If clumping was present, an additional Percoll clean-up step was undertaken. Nuclei were suspended in a wash buffer, and a manual cell count with C-chip (Trypan blue staining) was performed.

## Single nuclear RNA sequencing

The single-nuclei suspensions derived from each tumour were loaded onto separate channels of a Chromium 10× Genomics single cell 5′ version 2 library chip as per the manufacturer's protocol, aiming for 7000 nuclei per channel. Four 10× channels were loaded from each tumour sample. cDNA sequencing libraries were prepared as per the manufacturer's protocol and sequenced using an Illumina Hi-seq 4000.

## Quality control of single nuclear and single-cell data

Raw sequence reads in FASTQ format were processed and aligned to the GRCh38 version 1.2.0 human reference transcriptome through the Cellranger version v3.0.2 pipeline (10× Genomics) using default parameters. For murine and human mesangial cell data, expression matrices were obtained from previously published studies[4,24].

The expression matrices were processed with the SoupX package (V1.4.8) for R to estimate and remove cell-free mRNA contamination prior to analysis with the Seurat version 4.0.1 package for R[43,44].

Cells/nuclei with fewer than 1000 genes and greater than 7500 genes were filtered, as well as those in which mitochondrial genes represented 10% or greater of total gene expression.

Each run was processed with the Scrublet V0.2.2 pipeline using default parameters to obtain per-cell doublet scores. The standard Seurat processing pipeline was then performed on each sample up to the clustering stage, where an over-clustered manifold was produced (resolution parameter = 10), and any cell clusters with mean doublet score >0.1 were removed[45]. Subsequently, any remaining cells/nuclei called doublets were removed from the analysis (totally removed = 4113).

## Clustering of single nuclear and single-cell data

Data were processed using the Seurat package v4.0.1 for R[46]. To account for variations in the cell-cycle stage, Seurat's 'CellCycleScoring' function was performed on the remaining nuclei/cells to produce a quantitative estimation of the cell cycle stage. Log normalisation was then performed using the "NormalizeData" function with default parameters such that a total number of counts per cell was normalised to 10,000 prior to data scaling, which used cell cycle score, mitochondrial gene expression level and the total unique molecular identifiers (UMIs) per cell as regression variables. Variable features were identified using the "FindVariableFeatures" function to select for 2000 most variably expressed genes. Principal-component (PC) analysis was then performed on log-transformed data using the "RunPCA" function, and the optimum number of PCs for downstream analysis was identified using the "JackStraw" and "JackStrawPlot" functions. The neighbourhood graph was then computed using these and other default parameters, and the graph was embedded in two dimensions using uniform manifold approximation and projection. Clustering of data was performed by Louvain community detection on the neighbourhood graph with default resolution set to 1.

Pseudobulking of murine mesangial-like cells was performed by subsetting the cells by sample and then extracting the raw counts after QC filtering. Raw counts were then aggregated and exported for gene expression analysis.

## Annotation and gene expression for single nuclear and single-cell data

Annotation of clusters was performed by computing differentially expressed genes using the Wilcoxon rank sum test via Seurat's "FindMarkers" function, with genes requiring a minimum of 10% expression within a cluster to be returned and a log2FC threshold of 0.2. The adjusted p-value (post-Bonferroni correction) cut-off was set to 0.05. Cells/nuclei were annotated based on differential expression of genes previously identified in literature[4,24]. Cells/nuclei co-expressing REN, NRARP and NOTCH1 were identified using the "WhichCells" function, specifying log-normalised expression >0 of all three genes as a condition. Gene expression was visualised using the "FeaturePlot" function in Seurat and the ggplot2 package for R[42].

To calculate the ratio of NOTCH1 target to NRARP mRNA counts, per-nuclei log-normalised expression values of target genes expressed in the dataset were divided by the value for NRARP, producing a per-nuclei ratio value. To account for the technical bath effect, per-sample normalisation was performed against each sample's endothelial nuclei, which depend on NOTCH signalling for development[47]. Tumour nuclei ratios were divided by the median target gene/NRARP ratio value for endothelial nuclei in each sample. Results were visualised using the ggplot2 package in R[42].

## Reanalyses of published bulk transcriptomes

Raw sequencing reads for publicly available reninoma samples were downloaded from the Sequence Read Archive (SRA) (Accession: SRX535183) and mapped to GRCh37 (Gencode version 19) using STAR (version 2.7.10)[36]. Bulk transcriptomes were genotyped using the "matchBAMs" function in AlleleIntegrator[48]. RNA fusions were detected using STAR-fusion (version 1.12). Putative RNA fusions were assessed using junction read count and spanning fragments, then manually inspected using FusionInspector. Orthogonal evaluation of exon-level expression was performed using bcftools mpileup (version 1.9), bedGraphToBigWig (kentUtils update v302) and bwcat (version 1.5.2).

RNA-seq transcript quantification was performed using RSEM[49,50]. Sanger-reprocessed RNA counts were directly compared to downloaded counts from Gene Expression Omnibus (Supplementary Figure 4, Accession: GSE57401). For this analysis, microRNAs, genes with multiple isoforms and genes with 0 counts were omitted. Linear regression and the line of best fit were determined using the lm() function in R.

'Hotspot' variants affecting NOTCH1, NOTCH2, NOTCH3, NOTCH4 or JAG1 were downloaded from the COSMIC census[34] and searched for by manual inspection using the integrated genomics viewer (IGV). Hotspot mutations in other genes were also assessed (BRAF V600E,

*IDH1* R132R, *KRAS* G12/G13, *NRAS* Q61, *HRAS* G113R/Q61R, *TP53* R175H/R248Q/R273H and *PIK3CA* H1047R/E545K).

For Wilms tumour, renal cell carcinoma, congenital mesoblastic nephroma and normal kidneys, both in-house and publicly available raw expression matrices were downloaded[26–30]. All data were processed as detailed in the Bulk RNA analysis above.

## Reporting summary

Further information on research design is available in the Nature Portfolio Reporting Summary linked to this article.

## Data availability

The authors declare that all data supporting the findings of this study are available within the article, Source Data file and its Supplementary Information files or from the corresponding author. Sequencing data have been deposited at the European Genome-Phenome Archive (http://www.ebi.ac.uk/ega/), which is hosted by the European Bioinformatics Institute (accession numbers EGAD00001010888 (WGS), EGAD00001010889 (bulk RNA), EGAD00001010887 (single nuclear RNA). The data are available under restricted access due to data privacy laws, access may be granted following an application to the Data Access Committee, datasharing@sanger.ac.uk. Third-party data used within this study are available via the corresponding references. Re-analysed reninoma data is available GSE57401. Human and mouse mesangial data are available via EGA, EGAD00001008030. and via GEO, GSE160048. Bulk transcriptomes of other renal tumours GSE157256, GSE62944 and via EGA, EGAD00001008470, EGAS00001002487, EGAS00001002534 and EGAD00001004346. Source data are provided in this paper.

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

## Acknowledgements

This project was funded by The Little Princess Trust (grant number CCLGA 2019 27) and the Wellcome Trust Grants (grants 206194; 108413/A/15/D; 223135/Z/21/Z). G.M. was supported by La Ligue Contre le Cancer and by Assistance Publique Hôpitaux de Paris. P.J. is supported by a grant from the Wellcome Trust to the Wellcome Sanger Institute, 296194 and a Cancer Research UK Programme Grant, C609/A27326. We thank Professor Freddy Ratke for sharing his expertise on NOTCH1 and Professor Jan Danser for helping us access raw sequencing data of his reninoma study. We thank the clinical teams involved in the care of patients, in particular Professor Imran Mushtaq, Mrs Eileen Brennan and Dr Lauren Roe of Great Ormond Street Hospital for Children NHS Foundation Trust, London, UK. We are indebted to our patients and their families for participating in this research.

## Author contributions

S.B. conceived of the experiment and directed genomic analyses. T.D.T. and J.E.L. analysed the data, with contributions from N.D.A., T.H.H.C., H.L.S. and T.R.W.O. Samples were curated and/or experiments were performed by A.L., E.A. and R.A.S. P.J. provided expertise on NOTCH1. K.P.J. provided expertise on renal tumours of childhood and K.T. on reninoma. T.M. contributed renal cell carcinoma data. Pathological expertise was provided by G.M., J.C.H., D.R. and N.S. Statistical expertise was provided by M.DY. T.D.T., J.E.L. and S.B. wrote the paper. M.T., T.C. and S.B. co-directed the study.

## Competing interests

The authors declare no competing interests.
