## [Peer Review File · Nature Communications]

REVIEWER COMMENTS

Reviewer #1 (Remarks to the Author): Expert in cancer genomics, scRNA-seq, and tumour microenvironment

In the manuscript entitled “Brief Communication: Targetable dual hit NOTCH1 rearrangements in reninoma”, the authors report activating rearrangements of NOTCH1 in 2 cases of reninoma, very rare juxtaglomerular cell tumor. In addition to the genetic rearrangement, overexpression of NOTCH1 (C terminus) and selected downstream target genes corroborates activation of NOTCH signaling in the tumor cells. It is a clear well-written case report.

NOTCH signaling may play a key role in the maintenance and expansion of both normal mesangial-like cells and reninoma. It is not clear whether the reported genetic alterations are driver mutations.

Specific comments:

1. In Figure 2b: single cell RNA sequencing data of murine kidney cells, NOTCH1 signaling was presented to show high expression of NOTCH1 and NRARP in normal renin-expressing mesangial-like cells. Are there human data demonstrating similar results?
2. As suggested in Figure 2b, normal renin-expressing mesangial cells seem to express high levels of NOTCH1 and NRARP transcripts compared to other cell types in the kidney. Is it possible to estimate overexpression of the two transcripts in reninoma compared to the normal renin-expressing mesangial cells? In other words, is the overexpression in Fig 1 d and f due to the cellular expansion or de-regulation of the genes?

Reviewer #2 (Remarks to the Author): Expert in renal cancer genomics, scRNA-seq, and bioinformatics

The study by Treger and colleagues proposes a role for NOTCH1 truncation of the extracellular domain and NRARP 1-copy deletion in a rare kidney tumor Reninomas. The discovery may open new therapeutic opportunities for this rare tumor type. The weaknesses of the paper are very small sample size, inappropriate reference used to analyze consequences of NOTCH1 activation, and lack of experimental data supporting potential effectiveness of NOTCH1 inhibitor on reninomas.

There are several issues that should be address if the authors are invited to submit a revised manuscript.

1. Line 85, the authors should elaborate “more complex configuration”.

2. Figure 1a, case 2, it will be easier for the readers if the inversed exons are highlighted in some way.
3. Line 113/Figure 1d and line 120/Figure 2a, the caveat of using bulk normal kidney tissue as reference is that it does not provide an appropriate baseline for reinomas, since the tumor originates from juxtaglomerular cells which are rare and must be poorly represented in bulk kidney tissue. One possible strategy is to use pseudo-bulk profile of juxtaglomerular cells from human single cell data, and evaluate enrichment of NOTCH pathway in juxtaglomerular cells and reinomas using algorithms like ssGSEA/GSVA .
4. Line 138 Have the authors analyzed kidney single cell datasets of human instead of mouse to identify juxtaglomerular cells?
5. L144, figure 2d, it is preferred to have two plots showing expression of REN and NOTCH1 separately. It is meaningful to know if there are other cells express REN or NOTCH1.
6. L145, the authors should specify what comparison is made to conclude NRARP expression is up-regulated, and what statistical test is used.
7. L150, any data support “detection of the NOTCH1 intracellular domain using a specific epitope at the S3 cleavage site (D3B8 antibody) supports its release by γ -secretase cleavage in case 1”? it will be more convincing if using cell line derived from a reinomas patient to conduct experiments to support the proposal of treating reinomas patients with γ - secretase inhibitors. Or if γ - secretase inhibitors are successfully applied to other cancer types with NOTCH1 activation like T cell leukemia as mentioned in the manuscript.

Reviewer #3 (Remarks to the Author): Expert in reinomas

Treger et al. used whole genome sequencing to study 2 human reinomas, and come up with NOTCH1 and NRARP (its negative regulator) as potential players. Since NRARP is activated, its presumed negative effects are assumed to be unable to overrule NOTCH1 upregulation. There is no evidence for this concept. The authors might check the transcriptome analysis of human reinomas described by A Martini et al. (Hypertension 2017). How do these data compare to theirs, in particular with regard to Notch1/NRARP. In other words, after these 2 UK cases, would indeed this information be relevant for genomic assays worldwide, or does it represent a local phenomenon only. It is suggested that NOTCH1 is involved in renin release/synthesis. Here, the authors might compare their data to the data published by R Gomez et al. on this topic (e.g., on the genes that determine the identity of a renin cell: where does NOTCH1 fit in?). Somehow, the authors suggest to use NOTCH1 inhibitors, but we have no clue whether indeed they suppress renin: please provide evidence for this. Why would such drugs be better than a renin inhibitor (or surgical removal), also considering side effects?

Reviewer #4 (Remarks to the Author): Expert in renal cancer genomics and single-cell genomics

Summary

In the current manuscript, Treger et al. perform whole genome sequencing (WGS) and analyses on human reninomas (juxtaglomerular cell tumors), including primary tumors (n=2 samples from case 1; n=1 sample from case 2), metastasis (case 2), normal kidney (both cases) and blood cell derived DNA (case 1). The authors called all classes of somatic variation. The key finding was a 0.8 MB deletion on chromosome 9q seen in tumours of both patients harboring rearrangements that generated an activating truncation of the NOTCH1 gene while removing one copy of the NOTCH1 inhibitor, NRARP. Next, the authors Validated the results by RNA sequencing and Immunohistochemistry for the reninomas case. Furthermore, they also showed that the expression of NOTCH1 in tumour specimens derived from two patients with reninomas was higher than that of normal kidney tissues taken from the 2 cases and 119 normal kidney samples in 108 unmatched donors.

The authors not only studied some targets of the NOTCH1-NRARP pathways but also re-analyzed published transcriptomes of single murine kidney cells and got the desired results they expected. In this work, the discovery of somatic changes that define rare reninomas may have significant value in the translation to the clinic application. I think this paper is suitable for publication in Nature Communications. However, the following issues need to be addressed before the paper is entirely accepted:

1. The authors need to get some new samples from renal tumours, such as clear cell renal carcinoma (ccRCC), to be a positive control group in Figure 1d, since the evidence from negative control only is not enough to draw a meaningful conclusion.
2. The authors need to do the Ligand/Receptor interaction analysis in re-analyzed published transcriptomes of single murine kidney cells, finding more evidence to support the hypothesis that dual hit rearrangements targeting NOTCH1 and NRARP underpin reninomas.
3. In line 76 to 78, the paper states "We examined the genomes of primary tumours (n=2 samples from case 1; n=1 sample from case 2), metastasis (case 2), normal kidney (both cases) and blood cell derived DNA (case 1).", but I cannot see any results about the "blood cell derived DNA (case 1)." Meanwhile, the authors need to explain the reason for performing the two primary tumours sample in the same case.
4. In line 108, "Figure 1c" should be "Figure 1e".
5. In line 113, " Supplementary Figure 1" should be "Supplementary Figure 1c".

Manuscript NCOMMS-22-48009
Targetable *NOTCH1* rearrangements in reninoma

Response to Reviewers

Reviewer 1

#	Comment	Response
1.0	In the manuscript entitled “Brief Communication: Targetable dual hit NOTCH1 rearrangements in reninoma”, the authors report activating rearrangements of NOTCH1 in 2 cases of reninoma, very rare juxtaglomerular cell tumor. In addition to the genetic rearrangement, overexpression of NOTCH1 (C terminus) and selected downstream target genes corroborates activation of NOTCH signalling in the tumor cells. It is a clear well-written case report. NOTCH signalling may play a key role in the maintenance and expansion of both normal mesangial-like cells and reninoma	We thank the Reviewer for their summary of our manuscript and their helpful suggestions for improvement.
1.1	It is not clear whether the reported genetic alterations are driver mutations.	The NOTCH1 rearrangements we identified are canonical variants that have been described in numerous cancer types and which are the target of various drug development efforts. As a “Domain 1” mutation it would be considered a driver. For example, in the annotation system of the clinical whole genome sequencing service of cancers provided by NHS England, the canonical NOTCH1 rearrangement we identified would be annotated as a bonafide and clinically actionable driver event in a tumour agnostic way, akin to a BRAF V600E mutation in any tumour context (personal communication, Dr Patrick Tarpey, Lead Clinical Scientist of the NHS East of England Genomics Laboratory Hub).

		We have amended the manuscript to articulate this more clearly. Changes to manuscript: Page 3, lines 75-78 Page 4, lines 107-111
1.2	In Figure 2b: single cell RNA sequencing data of murine kidney cells, NOTCH1 signalling was presented to show high expression of NOTCH1 and NRARP in normal renin-expressing mesangial-like cells. Are there human data demonstrating similar results?	Thank you for raising this point. Since submission of this manuscript, we have published a paper containing a large data set of human kidney cells (Li et al., 2022). In this data we have been able to identify mesangial-like cells, as defined in the murine data set, and have used these for our analyses. And the answer is yes – mesangial-like human cells exhibit high expression of NOTCH1 and NRARP.  b, UMAP plots showing normalised expression of mesangial-like cell marker genes, NRARP and NOTCH1 across human and murine cells

		Changes to manuscript: 1) The above plot is shown in Supplementary Figure 3b.																												
1.3	As suggested in Figure 2b, normal renin-expressing mesangial cells seem to express high levels of NOTCH1 and NRARP transcripts compared to other cell types in the kidney. Is it possible to estimate overexpression of the two transcripts in reninoma compared to the normal renin-expressing mesangial cells? In other words, is the overexpression in Fig 1 d and f due to the cellular expansion or de-regulation of the genes?	This is a most interesting question. Given the known function of the NOTCH1 rearrangement to dysregulate (i.e. constitutively activate) NOTCH1 signalling, one would expect that in addition to an increase in cell number, NOTCH1 signalling is more active in each reninoma cell. We tackled the question head on by generating single cell transcriptomes of the two reninomas of our study by sequencing single nuclei derived from fresh frozen tissue using the Chromium10X platforms and standard protocols. This is a technique that our programme at Sanger, as one of the key contributors of the Human Cell Atlas project, is well versed in. With this data we addressed two questions: (i) Do tumour cells express more renin per cell than human juxtaglomerular cells? And the answer to the question is a resounding “yes”. b REN   <caption>Approximate data from the REN box plot</caption>   Cell Type n Median Q1 Q3 Min Max     PD54845a 3688 ~4.5 ~3.5 ~5.0 ~0.5 ~7.5   PD50642a 5989 ~3.0 ~2.5 ~4.0 ~0.5 ~6.0   Human MLC 350 ~1.5 ~1.0 ~2.0 ~0.5 ~4.0   	Cell Type	n	Median	Q1	Q3	Min	Max	PD54845a	3688	~4.5	~3.5	~5.0	~0.5	~7.5	PD50642a	5989	~3.0	~2.5	~4.0	~0.5	~6.0	Human MLC	350	~1.5	~1.0	~2.0	~0.5	~4.0
Cell Type	n	Median	Q1	Q3	Min	Max																								
PD54845a	3688	~4.5	~3.5	~5.0	~0.5	~7.5																								
PD50642a	5989	~3.0	~2.5	~4.0	~0.5	~6.0																								
Human MLC	350	~1.5	~1.0	~2.0	~0.5	~4.0																								

b, Box plot quantifying renin expression in tumour cells and mesangial like cells (MLC).

(ii) Is NOTCH1 signalling dysregulated in reninoma cells compared to human juxtaglomerular cells? To this end we measured on a per cell level, the number of *NOTCH1* effector transcripts per molecule of *NRARP*, and found a relative non-increase of *NRARP* mRNA in the presence of elevated NOTCH1 signalling.

c, Box plots showing ratio of *NOTCH1* target genes vs. *NRARP* for tumours and mesangial like cells (MLC)

d, Box plot quantifying NOTCH1 expression in tumour cells and mesangial like cells (MLC).

Overall these analysis would suggest that reninoma cells produce more renin per cell and display evidence of dysregulated NOTCH1 signalling.

Changes to manuscript:

- Description of additional experiments and findings in main text, Page 5, lines 142-167
- Addition of Figure 3b, 3c, 4d
- Additional methods added to methods section 463-532

Reviewer 2

#	Comment	Response
2.1	The study by Treger and colleagues proposes a role for NOTCH1 truncation of the extracellular domain and NRARP 1-copy deletion in a rare kidney tumor Reninomas. The discovery may open new therapeutic opportunities for this rare tumor type. The weaknesses of the paper are very small sample size, inappropriate reference used to analyze consequences of NOTCH1 activation, and lack of experimental data supporting potential effectiveness of NOTCH1 inhibitor on reninomas.	We thank the Reviewer for their helpful suggestions. We have addressed specific comments below and would like to respond here on the sample size issue. We entirely agree and would be most keen to study more tumours. An invariable shortcoming of our study is its sample size; reninomas are really very rare. In what must be one of the world's largest collection of renal tumours - the archives of the Royal Free Hospital (one of Europe's largest kidney tumour centres) and of Great Ormond Street Hospital – there were only two cases available. This situation is unlikely to change over the next few years, and we would suggest that our findings are too important to remain unpublished; they may help patients and they provide insights into a rare, physiologically important cell type. The Reviewer may be pleased to hear that for this revision we re-analysed bulk transcriptome data derived from four fresh frozen reninomas that Reviewer 3 had pointed us to (Martini et al., 2017). We contacted the senior author of the study in question, Jan Danser, who kindly signposted us to the raw RNAseq data which we downloaded and processed to be able to look for gene fusions. We also regenerated expression count tables which perfectly correlated with the published expression counts (Supplementary Figure 4). We performed three analyses: (i) First, we compared gene expression by looking for the overlap in differentially expressed genes which, amazingly, contained immediate canonical NOTCH1 effectors and even NOTCH1 and NRARP amongst the shared differentially expressed genes.

		(ii) Encouraged by this finding, we then looked for fusions events algorithmically and manually in NOTCH1 and its paralogues but did not find any. (iii) We then looked for hotspot mutations and found the NOTCH1 T349P variant in all four cases. This variant has been reported as a driver in a subset of ALCL (anaplastic large cell lymphoma) and has been functionally evaluated in the original study and in one further study since (Larose et al., 2021; Pennarubia et al., 2022). However, in the absence of DNA sequences to be able to validate the variant as genuine and as somatic, we hesitate to report it. Still, the expression analyses enables us to corroborate a role for Notch signalling in reninoma which we report in a dedicated paragraph and figure. Changes to manuscript:  1) Paragraph reporting the finding (Page 6, lines 170-193) 2) New figure (Figure 4) 3) Corresponding methods (Lines 534-552) 4) Supplement (Supplementary Figure 4 and Supplementary Table 8).
2.2	Line 85, the authors should elaborate “more complex configuration”.	We have changed the manuscript according to this suggestion: Page 4, lines 105-107.
2.3	Figure 1a, case 2, it will be easier for the readers if the inversed exons are highlighted in some way.	Thank you, this has been done.
2.4	Line 113/Figure 1d and line 120/Figure 2a, the caveat of using bulk normal kidney tissue as reference is that it does not provide an appropriate baseline for reninomas, since the tumor originates from juxtaglomerular cells which are rare and must be poorly represented in bulk kidney tissue. One possible strategy is to use pseudo-bulk profile of	We tackled this weakness head on in three ways. i) We generated pseudo-bulks from human and murine mesangial like cells, which we used as references for downstream analyses of bulk RNA data (differential gene analyses and gene expression).

juxtaglomerular cells from human single cell data, and evaluate enrichment of NOTCH pathway in juxtaglomerular cells and reninomas using algorithms like ssGSEA/GSVA .

c, *NOTCH1* expression (log-cpm (counts per million)) in reninomas, compared to mesangial like cells (psuedobulks, n=2), normal kidney (n=332), congenital mesoblastic nephroma (CMN, n=21), Wilms tumour (n=308) and renal cell carcinoma (RCC, n = 824).

ii) We generated single nuclear data from the two fresh frozen tumours in our study to enable exact quantitative analyses, to replace the bulk analyses the Reviewer questioned. In addition, we now have acquired human, rather than murine, mesangial-like cell data from a data set we have published since submission of this manuscript (Li et al., 2022).

With this data we addressed two questions:

(ii.a) Do tumour cells express more renin per cell than human juxtaglomerular cells? And the answer to the question is a resounding “yes”.

b *REN*

b, Box plot quantifying renin expression in tumour cells and mesangial like cells (MLC).

(ii.b) Is NOTCH1 signalling dysregulated in reninoma cells compared to human juxtaglomerular cells? To this end we measured on a per cell level, the number of *NOTCH1* effector transcripts per molecule of *NRARP*, and found a relative non-increase of *NRARP* mRNA in the presence of elevated NOTCH1 signalling.

c, Box plots showing ratio of *NOTCH1* target genes vs. *NRARP* for tumours and mesangial like cells (MLC)

Overall these analysis would suggest that reninoma cells produce more renin per cell and display dysregulated *NOTCH1* signalling.

iii) In a second new analysis, we assessed domain specific expression of *NOTCH1* comparing, in bulk transcriptome data, mutant *NOTCH1* (from reninoma tumours) with wildtype *NOTCH1* (normal tissues and other renal tumours). This beautifully shows just how aberrant *NOTCH1* expression is in tumours.

		b NOTCH1  Intracellular domain Transmembrane domain Extracellular domain Ratio of read coverage of ICD:ECD y=1 ● PD50642a ○ PD54845a CMN n=21 normal kidney n=13 RCC n=39 reninoma n=4 Wilms n=16 Changes to manuscript:  • Description of additional experiments and findings in main text, Page 4 130-132; Page 5, lines 142-167 • Addition of Figure 3b, 3c, 4d • Additional methods added to methods section 463-532 • New Figures 2b and 4c replacing previous Figure 1d
2.5	Line 138 Have the authors analyzed kidney single cell datasets of human instead of mouse to identify juxtaglomerular cells?	Yes – Please see our response to your previous point, 2.4.
2.6	L144, figure 2d, it is preferred to have two plots showing expression of REN and NOTCH1 separately. It is meaningful to know if there are other cells express REN or NOTCH1.	We have shown REN, NOTCH1 and NRARP expression for the murine cells and the new data set of human mesangial-like cells in a new supplementary figure 3b.

		b  b, UMAP plots showing normalised expression of mesangial-like cell marker genes, NRARP and NOTCH1 across human and murine cells
2.7	L145, the authors should specify what comparison is made to conclude NRARP expression is up-regulated, and what statistical test is used.	The statistical test we had used was Fisher’s exact test (two-sided). However, the analysis in question has superseded with new single nuclei analysis in the Revision. We describe our analytical approach including statistical evaluation in the methods section.
2.8	L150, any data support “detection of the NOTCH1 intracellular domain using a specific epitope at the S3 cleavage site (D3B8 antibody) supports its release by γ -secretase cleavage in case 1”? it will be more convincing if using cell line derived from a reinomas patient to conduct experiments to support the proposal of treating reinomas patients with γ - secretase inhibitors. Or if γ - secretase	We thank the Reviewer for raising this point. It will not be possible to generate cell lines from snap frozen tissues, and we may be waiting many years before we will get access to fresh tumour material again. However, we do know a great deal about the NOTCH1 rearrangement from the literature. As the Reviewer’s suggests, the NOTCH1 rearrangements we have found are canonical oncogenic NOTCH1 driver events that have been

inhibitors are successfully applied to other cancer types with NOTCH1 activation like T cell leukemia as mentioned in the manuscript.

extensively described in different tumour types and that are a “hot” target in some, especially in T-ALL. A variety of agents targeting NOTCH1 are under development and some have been tried in clinical studies [(Allen & Maillard, 2021) for review]. We stressed the preservation of the gamma-secretase cleavage site because some variants of *NOTCH1* rearrangements unfortunately lose this domain and thus become essentially untargetable (although there are efforts under way to target downstream transcription factor binding; see Freddy Radtke’s work). A case in point is a recent report by Zhang *et al* who successfully treated a child with *NOTCH1* rearranged metastatic glomus tumours with NOTCH1 inhibitors (Zhang et al., 2022).

As for targeting NOTCH1 in meningioma, we would not suggest to consider this as a replacement of surgery. However, in individuals in whom surgery is not possible and blood pressure control is intractable we would consider targeting mutations. In a clinical context, the *NOTCH1* rearrangement would be reported as a Domain 1 driver, i.e. a bonafide driver event considered to be targetable. Therefore, it would be a target that is on the table.

The Reviewer’s comments clearly highlight that we failed to adequately discuss both points. We have therefore amended the manuscript accordingly.

Changes to manuscript:

Page 4, lines 107-114

Page 7, lines 215 - 222

Reviewer 3

#	Comment	Response
3.1	Treger et al. used whole genome sequencing to study 2 human reninomas, and come up with NOTCH1 and NRARP (its negative regulator) as potential players.	We thank the Reviewer for their helpful comments.
3.2	Since NRARP is activated, its presumed negative effects are assumed to be unable to overrule NOTCH1 upregulation. There is no evidence for this concept.	We apologise for any overstatement regarding the role of NRARP. To address the Reviewer's point, we generated single cell transcriptomes of the two reninomas of our study by sequencing single nuclei derived from fresh frozen tissue using the Chromium10X platforms and standard protocols. This is a technique that our programme at Sanger, as one of the key contributors of the Human Cell Atlas project, is well versed in. With this data we addressed the question by measuring on a per cell level, the number of NOTCH1 effector transcripts per molecule of NRARP, and found a relative (and significant) non-increase of NRARP mRNA in the presence of elevated NOTCH1 signalling.

		C NRARP > NOTCH1 targets NOTCH1 targets > NRARP HEYL $p < 10^{-4}$ $p < 10^{-6}$ HEY2 $p < 0.0002$ $p < 10^{-13}$ HEY1 $p < 10^{-7}$ $p < 10^{-18}$ HES5 $p < 10^{-11}$ $p < 10^{-9}$ HES4 $p < 0.001$ HES1 $p < 0.02$ $p < 0.0001$ Ratio of NOTCH1 target over NRARP mRNA counts ■ PD50642a □ PD54845a ▒ Human MLC c, Box plots showing ratio of NOTCH1 target genes vs. NRARP for tumours and mesangial like cells (MLC) However, even with this data we would agree that our excitement over the NRARP truncation was too stark, and we have throughout the manuscript toned down the NRARP finding. In particular, we have taken the “dual hit” notion out of the title.
3.3	The authors might check the transcriptome analysis of human reninomas described by A Martini et al. (Hypertension 2017). How do these data compare to theirs, in particular with regard to Notch1/NRARP.	Thank you for highlighting this reference which we had failed to identify in our initial review of the literature. Martini et al analysed gene expression of 4 reninoma (7 samples) using RNA sequencing of fresh, frozen material. Unfortunately, the authors did not look for gene fusions and did not perform DNA sequencing

		We contacted the senior author of the study in question, Jan Danser, who kindly signposted us to the raw RNAseq data which we downloaded and processed to be able to look for gene fusions. We also regenerated expression count tables which perfectly correlated with the published expression counts. We performed three analyses: (i) First, we compared gene expression by looking for the overlap in differentially expressed genes which, amazingly, contained immediate canonical NOTCH1 effectors and even NOTCH1 and NRARP amongst the shared differentially expressed genes. (ii) Encouraged by this finding, we then looked for fusions events algorithmically and manually in NOTCH1 and its paralogues but did not find any. (iii) We then looked for hotspot mutations and found the NOTCH1 T349P variant in all four cases. This variant has been reported as a driver in a subset of ALCL (anaplastic large cell lymphoma) and has been functionally evaluated in the original study and in one further study since (Larose et al., 2021; Pennarubia et al., 2022). However, in the absence of DNA sequences to be able to validate the variant as genuine and as somatic, we hesitate to report it. Still, the expression analyses enables us to corroborate a role for Notch signalling in reinoma which we report in a dedicated paragraph and figure. Changes to manuscript: 1) Paragraph reporting the finding (Page 6, lines 170-193)2) New figure (Figure 4)3) Corresponding methods (Lines 534-552)
--	--	--

		4) Supplement (Supplementary Figure 4 and Supplementary Table 8).
3.4	In other words, after these 2 UK cases, would indeed this information be relevant for genomic assays worldwide, or does it represent a local phenomenon only.	This is perhaps a point that we both agree and disagree with. Unfortunately, to date no one else has performed unbiased DNA WGS to define mutations of reninoma so our two cases is all we have. It is not inconceivable that we may have sequenced two exceptions with NOTCH1 rearrangements which would, however, be an extraordinary co-incidence. More likely, as has been described in glomus tumours, as more reninomas will be sequenced, we may find a variety of driver events that functionally converge in aberrant Notch signalling. The Reviewers thought clearly highlights that we ought to further discuss the limitations of our study which we have done in this Revision (see page 7, lines 204-208).
3.5	It is suggested that NOTCH1 is involved in renin release/synthesis. Here, the authors might compare their data to the data published by R Gomez et al. on this topic (e.g., on the genes that determine the identity of a renin cell: where does NOTCH1 fit in?).	We thank the Reviewer for raising this point and apologise for lack of discussion in the manuscript. In our manuscript we put forward this hypothesis (clearly marked as a hypothesis). There is pre-existing evidence to support this notion, including:  1) A mouse study showed that Notch1 signalling controls the genetic program that confers the dual endocrine–contractile phenotype of the juxtaglomerular cell (Castellanos-Rivera et al., 2015). 2) Rbpj knock out mice (RBPJ is the classical downstream effector of NOTCH1 signalling, promoter binding of which is used as a reporter by NOTCH1 physiologists) displayed a significant reduction in the number of renin-positive juxtaglomerular apparatuses (JGA) and a reduction of renin secretion in cells of the JGA (Rivera et al., 2011). 3) A study of acute kidney injury in mice showed a connection between NOTCH1 signalling and the renin-angiotensin system, and that inhibition of NOTCH1 signalling lead to a reduction in the activity of the renin-angiotensin system (Wyss et al., 2018).

		4) A study in rat shows direct binding of the Notch1 intracellular domain to the renin promoter (Pan et al., 2005). It would therefore seem plausible to suggest a link between NOTCH1 and renin secretion. As our initial manuscript had been conceived as a Brief Report for a different Nature journal, we were unable to discuss these references in detail. In the new Nature Communications format we are no longer constrained, and are now able to discuss these studies in detail. Clearly, as the Reviewer's comment shows, our previous discussion was too brief. Changes to manuscript: Page 3, lines 72-75
3.6	Somehow, the authors suggest to use NOTCH1 inhibitors, but we have no clue whether indeed they suppress renin: please provide evidence for this. Why would such drugs be better than a renin inhibitor (or surgical removal), also considering side effects?	We empathically agree with the Reviewer, as stated in our manuscript: “Moreover, in individuals in whom surgical resection is contra-indicated, it may be reasonable to consider trialling NOTCH1 inhibitors as a medical treatment of reninoma.” We did not suggest to trial a NOTCH1 inhibitor as first line treatment of reninoma. If we were to encounter a patient with no other treatment options (inoperable disease, intractable hypertension), we would, however, consider a NOTCH1 inhibitor. To avoid any ambiguity, we have modified the above sentence to: “Our findings have immediate clinical relevance. Reninoma is primarily a surgical disease, even in cases of metastases. However, occasionally surgery will be contra-indicated or technically unfeasible. In this context, as reninomas do not respond to conventional anti-cancer therapies (i.e. cytotoxic agents or ionising radiation), the NOTCH1 rearrangement we describe here would be considered a targetable mutation. It may be

		reasonable to consider trialling NOTCH1 inhibitors in patients with otherwise incurable reninoma. Interestingly, using such a precision oncology approach, Zhang et al recently reported successful treatment with NOTCH1 inhibitors of a child with NOTCH1 rearranged metastatic glomus tumours.”(Zhang et al., 2022)
--	--	---

Reviewer 4

#	Comment	Response
4.1	In the current manuscript, Treger et al. perform whole genome sequencing (WGS) and analyses on human reninomas (juxtglomerular cell tumors), including primary tumors (n=2 samples from case 1; n=1 sample from case 2), metastasis (case 2), normal kidney (both cases) and blood cell derived DNA (case 1). The authors called all classes of somatic variation. The key finding was a 0.8 MB deletion on chromosome 9q seen in tumours of both patients harboring rearrangements that generated an activating truncation of the NOTCH1 gene while removing one copy of the NOTCH1 inhibitor, NRARP. Next, the authors Validated the results by RNA sequencing and Immunohistochemistry for the reninomas case. Furthermore, they also showed that the expression of NOTCH1 in tumour specimens derived from two patients with reninomas was higher than that of normal kidney tissues taken from the 2 cases and 119 normal kidney samples in 108 unmatched donors. The authors not only studied some targets of the NOTCH1-NRARP pathways but also re-analyzed published transcriptomes of single murine kidney cells and got the desired results they expected. In this work, the discovery of	We thank the Reviewer for their summary of our manuscript and their helpful suggestions for improvement.

	somatic changes that define rare reninomas may have significant value in the translation to the clinic application. I think this paper is suitable for publication in Nature Communications. However, the following issues need to be addressed before the paper is entirely accepted:	
4.2	The authors need to get some new samples from renal tumours, such as clear cell renal carcinoma (ccRCC), to be a positive control group in Figure 1d, since the evidence from negative control only is not enough to draw a meaningful conclusion.	We thank the Reviewer for raising this point. We have tackled this head on generating single nuclei data from the tumours by sequencing single nuclei derived from fresh frozen tissue using the Chromium10X platforms and standard protocols. This enabled us to perform direct comparisons between tumour cells and mesangial-like cells. Please note that we have replaced the murine data with human data which has become available since submission of this manuscript. In a new analysis, we assessed domain specific expression of NOTCH1 comparing, in bulk transcriptome data, mutant NOTCH1 (from reninoma tumours) with wildtype NOTCH1 (normal tissues and other renal tumours, as per the Reviewer's helpful suggestion). This beautifully shows just how aberrant NOTCH1 expression is in tumours.

b

b, Ratio of normalised coverage of *NOTCH1* intracellular signalling domain vs. extracellular domain, plotted for reninomas, congenital mesoblastic nephroma (CMN, n=21), Wilms tumour (n=16), normal kidney (n=14) and renal cell carcinoma (RCC, n = 39)

In a second new analysis, we generated a pseudo-bulk of murine and human mesangial-like cells. We compared *NOTCH1* expression in these pseudobulks, our in house reninoma data (n=4), published reninomas (n=7) and, as per the Reviewer's suggestion, 3 different groups of renal tumours (RCC (n=824), Wilms tumour (n=308) and CMN (n=21)).

		C  c, NOTCH1 expression (log-cpm (counts per million)) in reninomas, compared to mesangial like cells (pseudobulks, n=2), normal kidney (n=332), congenital mesoblastic nephroma (CMN, n=21), Wilms tumour (n=308) and renal cell carcinoma (RCC, n=824). Changes to manuscript:  • Description of additional experiments and findings in main text Page 4 130-132; Page 5, lines 142-167, Page 6, 187-193 • Addition of Figure 3b and 3c, Addition of Figure 4c • New Figures 2b and 4c replacing previous Figure 1d
4.3	The authors need to do the Ligand/Receptor interaction analysis in re-analyzed published transcriptomes of single murine kidney cells, finding more evidence to support the hypothesis that dual hit rearrangements targeting NOTCH1 and NRARP underpin reninomas.	Thank you for this suggestion. Whilst there are no tools to interrogate interactions within individual cells, what we have done in this revision is to use the single nuclei / cell data to precisely quantify gene expression.

i) We generated single nuclear data from the two fresh frozen tumours in our study to enable exact quantitative analyses. In addition, we now have acquired human, rather than murine, mesangial-like cell data from a data set we have published since submission of this manuscript.

With this data we addressed two questions:

(i.a) Do tumour cells express more renin per cell than human juxtaglomerular cells? And the answer to the question is a resounding “yes”.

b *REN*

b, Box plot quantifying renin expression in tumour cells and mesangial like cells (MLC).

(i.b) Is NOTCH1 signalling dysregulated in reninoma cells compared to human juxtaglomerular cells? To this end we measured on a per cell level, the number of *NOTCH1* effector transcripts per molecule of *NRARP*, and found a relative non-increase of *NRARP* mRNA in the presence of elevated NOTCH1 signalling.

c, Box plots showing ratio of *NOTCH1* target genes vs. *NRARP* for tumours and mesangial like cells (MLC)

Overall these analysis would suggest that reninoma cells produce more renin per cell and display dysregulated *NOTCH1* signalling.

Changes to manuscript:

- Description of additional experiments and findings in main text, Page 5, lines 142-167
- Addition of Figure 3b and 3c
- Additional methods added to methods section 463-532

4.4	In line 76 to 78, the paper states "We examined the genomes of primary tumours (n=2 samples from case 1; n=1 sample from case 2), metastasis (case 2), normal kidney (both cases) and blood cell derived DNA (case 1).", but I cannot see any results about the "blood cell derived DNA (case 1)." Meanwhile, the authors need to explain the reason for performing the two primary tumours sample in the same case.	Apologies for the confusion. As a germline control we had blood derived DNA available from case 1 and normal kidney from case 2. We did not identify a predisposition in either case. We have changed the manuscript to clarify this point: Page 3, lines 99-100 The reason for analysing two primary tumour samples from case 1 was that it was sampled as part of a national biobanking study of childhood renal tumours (UMBRELLA study) which advocates multi-site sampling specimens. As we had tissues from two different regions available, we sequenced both.
4.5	In line 108, "Figure 1c" should be "Figure 1e".	Thank you for pointing out this mistake, it has been amended.
4.6	In line 113, " Supplementary Figure 1" should be "Supplementary Figure 1c".	Thank you for pointing out this mistake, it has been amended.

References for Rebuttal

- Allen, F., & Maillard, I. (2021). Therapeutic Targeting of Notch Signaling: From Cancer to Inflammatory Disorders. *Frontiers in Cell and Developmental Biology*, 9, 1262. <https://doi.org/10.3389/FCELL.2021.649205/BIBTEX>
- Castellanos-Rivera, R. M., Pentz, E. S., Lin, E., Gross, K. W., Medrano, S., Yu, J., Sequeira-Lopez, M. L. S., & Gomez, R. A. (2015). Recombination signal binding protein for Ig-κJ region regulates juxtaglomerular cell phenotype by activating the myo-endocrine program and suppressing ectopic gene expression. *Journal of the American Society of Nephrology : JASN*, 26(1), 67–80. <https://doi.org/10.1681/ASN.2013101045>
- Larose, H., Prokoph, N., Matthews, J. D., Schleder, M., Högl, S., Alsulami, A. F., Ducray, S. P., Nuglozeh, E., Fazaludeen, M. F., Elmouna, A., Ceccon, M., Mologni, L., Gambacorti-Passerini, C., Hoefler, G., Lobello, C., Pospisilova, S., Janikova, A., Woessmann, W., Damm-Welk, C., ... Turner, S. D. (2021). Whole Exome Sequencing reveals NOTCH1 mutations in anaplastic large cell lymphoma and points to Notch both as a key pathway and a potential therapeutic target. *Haematologica*, 106(6), 1693–1704. <https://doi.org/10.3324/HAEMATOL.2019.238766>

- Li, R., Ferdinand, J. R., Loudon, K. W., Bowyer, G. S., Laidlaw, S., Muyas, F., Mamanova, L., Neves, J. B., Bolt, L., Fasouli, E. S., Lawson, A. R. J., Young, M. D., Hooks, Y., Oliver, T. R. W., Butler, T. M., Armitage, J. N., Aho, T., Riddick, A. C. P., Gnanapragasam, V., ... Mitchell, T. J. (2022). Mapping single-cell transcriptomes in the intra-tumoral and associated territories of kidney cancer. *Cancer Cell*, 40(12), 1583-1599.e10. <https://doi.org/10.1016/J.CCELL.2022.11.001>
- Martini, A. G., Xa, L. K., Lacombe, M. J., Blanchet-Cohen, A., Mercure, C., Haibe-Kains, B., Friesema, E. C. H., Van Den Meiracker, A. H., Gross, K. W., Azizi, M., Corvol, P., Nguyen, G., Reudelhuber, T. L., & Danser, A. H. J. (2017). Transcriptome Analysis of Human Reninomas as an Approach to Understanding Juxtaglomerular Cell Biology. *Hypertension*, 69(6), 1145–1155. <https://doi.org/10.1161/HYPERTENSIONAHA.117.09179>
- Pan, L., Glenn, S. T., Jones, C. A., & Gross, K. W. (2005). Activation of the rat renin promoter by HOXD10.PBX1b.PREP1, Ets-1, and the intracellular domain of notch. *The Journal of Biological Chemistry*, 280(21), 20860–20866. <https://doi.org/10.1074/JBC.M414618200>
- Pennarubia, F., Ito, A., Takeuchi, M., & Haltiwanger, R. S. (2022). Cancer-associated Notch receptor variants lead to O-fucosylation defects that deregulate Notch signaling. *The Journal of Biological Chemistry*, 298(12), 102616. <https://doi.org/10.1016/J.JBC.2022.102616>
- Rivera, R. M. C., Monteagudo, M. C., Pentz, E. S., Glenn, S. T., Gross, K. W., Carretero, O., Sequeira-Lopez, M. L. S., & Ariel Gomez, R. (2011). Transcriptional regulator RBP-J regulates the number and plasticity of renin cells. *Physiological Genomics*, 43(17), 1021–1028. <https://doi.org/10.1152/PHYSIOLGENOMICS.00061.2011>
- Wyss, J. C., Kumar, R., Mikulic, J., Schneider, M., Aebi, J. D., Juillerat-Jeanneret, L., & Golshayan, D. (2018). Targeted γ -secretase inhibition of notch signaling activation in acute renal injury. *American Journal of Physiology - Renal Physiology*, 314(5), F736–F746. <https://doi.org/10.1152/AJPRENAL.00414.2016/ASSET/IMAGES/LARGE/ZH20111783670007.JPEG>
- Zhang, E., Miller, A., Clinton, C., DeSmith, K., Voss, S. D., Aster, J. C., Church, A. J., Rahbar, R., Eberhart, N., Janeway, K. A., & DuBois, S. G. (2022). Gamma Secretase Inhibition for a Child With Metastatic Glomus Tumor and Activated NOTCH1. *https://Doi.Org/10.1200/PO.22.00099*, 6. <https://doi.org/10.1200/PO.22.00099>

REVIEWERS' COMMENTS

Reviewer #1 (Remarks to the Author):

My review comments are well addressed. For the small sample size, although could not be changed, the authors made great effort to compensate for it, using public sources.

Reviewer #2 (Remarks to the Author):

The authors have addressed my comments. I only have questions about Fig 4C in the revised manuscript.

1. What does the y-axis lcpm mean?
2. The four boxes should be colored or labeled on the x-axis to indicate which is which.
3. Are the positions of the dots random?

Reviewer #3 (Remarks to the Author):

All issues have been adequately addressed.

Reviewer #4 (Remarks to the Author):

The authors have addressed my main concerns and I have no more questions to ask.

Manuscript NCOMMS-22-48009
Targetable *NOTCH1* rearrangements in meningioma

Response to Reviewers

We would like to thank all the Reviewers for their most helpful comments during the revision process.

Reviewer 1

#	Comment	Response
1.0	My review comments are well addressed. For the small sample size, although could not be changed, the authors made great effort to compensate for it, using public sources.	We thank the Reviewer for their comments.

Reviewer 2

#	Comment	Response
1.0	The authors have addressed my comments. I only have questions about Fig 4C in the revised manuscript. 1. What does the y-axis lcpm mean? 2. The four boxes should be colored or labeled on the x-axis to indicate which is which. 3. Are the positions of the dots random?	We thank the Reviewer for their helpful suggestions for improvement. Regarding 4C 1. lcpm has been changed to logCPM (log counts per million), this metric was used in order to account for differences in library sizes. 2. The boxplots have been labelled with their respective tumour/normal group. 3. The dots have been jittered horizontally in order to visualise them individually.

Reviewer 3

#	Comment	Response
1.0	All issues have been adequately addressed.	We thank the Reviewer for their comments.

Reviewer 4

#	Comment	Response
1.0	The authors have addressed my main concerns and I have no more questions to ask.	We thank the Reviewer for their comments.